# Ensuring Physicochemical Fidelity of Generated Polymers with PoGE

## Abstract

Recent advances in machine learning have accelerated progress in chemistry, enabling new capabilities in molecular design, property prediction, and materials discovery. A critical challenge in materials science is designing polymers with targeted macroscopic properties. However, prior generative models often fail to produce chemically valid polymer structures, hindering progress toward this goal. We introduce PoGE (Polymer Generation and Evaluation), a framework comprising two complementary components: a physics-informed evaluation suite for polymer generative models, and an unconditional transformer-based generative model adapted to polymer representations. Building upon and extending established molecule-centric benchmarks, our evaluation quantifies the alignment between the generated and experimental property distributions using the Wasserstein distance. The generative model is trained on a hybrid corpus of synthetic and experimental polymer representations and enforces polymer-specific validity constraints ("p-validity") beyond the standard small-molecule validity. PoGE achieves high p-validity and significantly improved agreement with experimental property distributions compared to prior methods, even without explicit property conditioning during generation. By releasing a comprehensive benchmark, a high-quality pre-training corpus, and the trained model, PoGE establishes a foundation for conditional polymer generation tasks (e.g., on-demand reverse design), enabling targeted property optimization and accelerating reproducible, domain-aware polymer discovery.

## 1 Introduction

The rapid advancement of machine learning and artificial intelligence has revolutionized materials discovery, and generative models are emerging as powerful tools to design novel materials with tailored properties. Among these approaches, transformer-based language models have shown remarkable success in generating molecular structures by treating chemical representations (i.e. SMILES) as sequential data (Arús-Pous et al. (2020); Bagal et al. (2022); Pang et al. (2024); Strandgaard et al. (2025)). However, the application of these generative models to polymer design presents unique challenges that are not adequately addressed by existing molecular generation frameworks, particularly for on-demand reverse design, where researchers seek to synthesize polymers with specific macroscopic properties (e.g., high glass transition temperature or gas permeability).

The fundamental difference between small molecules and polymers lies in the availability and nature of training data. Generative models for drug-like molecules benefit from large publicly accessible databases such as PubChem (Kim et al. (2024)), ChEMBL (Mendez et al. (2019)), and ZINC (Irwin & Shoichet (2005)), which collectively contain millions of well-characterized compounds. These extensive datasets enable effective training and fine-tuning of models for property-driven molecular generation. In contrast, the polymer domain suffers from a scarcity of large open-access datasets (Mahmood et al. (2021); Zhang et al. (2024); Ross et al. (2025)). For homopolymers, publicly available databases, such as PolyInfo (Otsuka et al. (2011)), do not exceed a few tens of thousands of entries, limiting the direct application of data-hungry generative models developed for small molecules. Consequently, training generative models for polymers often requires the construction of large synthetic corpora for pre-training, followed by fine-tuning on curated experimental datasets to capture realistic polymer chemistry.

Compounding this data scarcity is a critical misalignment in evaluation. Standard molecular generation evaluation frameworks (e.g. MOSES (Polykovskiy et al. (2020)), guacaMOL (Brown et al. (2019))), designed for small molecules, fail to validate critical polymer-specific structural requirements such as valid polymerization motifs.

We introduce PoGE (Polymer Generation and Evaluation), which solves this by: (1) training an unconditional generative model for polymer structures, and (2) replacing molecule-centric benchmarks with physics-driven evaluation via Wasserstein distance (Panaretos & Zemel (2019)) between descriptor distributions. PoGE enables reliable on-demand reverse design by directly mapping macroscopic property targets to chemically plausible structures.

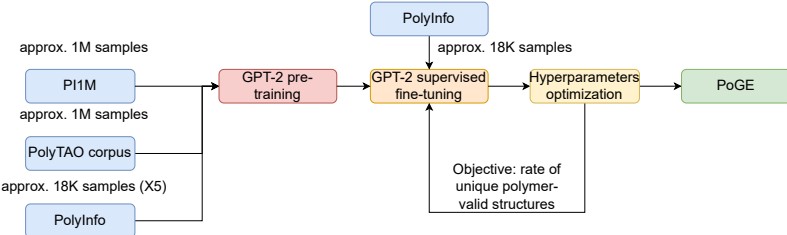

Figure 1: Scheme of PoGE training procedure.

## 2 RELATED WORK

Existing unconditional polymer generative models are basically limited to PI1M (Ma & Luo (2020)) - an unconditional generator, processing p-SMILES (a polymer-specific SMILES variant where '*' denotes the endpoints of the polymer repeat unit) using the recurrent neural network (RNN) architecture with gated recurrent units. The corpus of about 1 million synthetic polymer structures allowed machine learning discoveries of novel polymers with high gas permeability (Chen et al. (2024)) and conductivity (Khajeh et al. (2025)), as well as conditional generation frameworks that use structural descriptors as inputs (Qiu & Sun (2024)) (e.g., molecular weight, number of rings in monomer, see A.2.1 for detailed view). However, the generative model underlying PI1M is inherently limited by the relatively small initial training set (12,000 in the original work) and the nature of RNNs, which can struggle to model long-range dependencies and complex structural motifs characteristic of polymers.

While recent advances in conditional generation for polymer design, such as the PolyTAO model (Qiu & Sun (2024)), demonstrate the utility of using structural descriptors as constraints, this approach presents a significant practical limitation. The selection of these low-level features as inputs does not align with the intuitive goals of materials design. In practice, a researcher is far more likely to know the target macroscopic properties of a polymer (e.g., glass transition temperature, tensile strength) than the precise count of its underlying structural descriptors. Consequently, conditioning on such specific, atomic-level features, while computationally effective, creates a barrier for direct, application-driven discovery. Moreover, PolyTAO relies on PI1M as its training corpus, inheriting its distributional biases (Table 2).

A critical challenge in polymer generation frameworks extends beyond limited training data to the evaluation of output quality. Established benchmarks like MOSES (Polykovskiy et al. (2020)) and guacaMOL (Brown et al. (2019)) rely on Extended Connectivity Fingerprints (ECFP) (Rogers & Hahn (2010)), RDKit's SMILES-to-MOL conversion for validity assessments, and chemical heuristics embedded in BRICS fragments (Degen et al. (2008)) and Bemis–Murcko scaffolds (Bemis & Murcko (1996)) (see A.1 for more details). While these approaches are suitable for drug-like molecules, they fail to capture the intricate structural requirements of polymeric materials. For example:

- p-SMILES "[*]=CC[*]" passes RDKit (Landrum (2013)) validation but encodes an impossible polymer structure due to mismatched valency at the endpoints of the polymer repeat unit

- Synthetic Accessibility Score (SAscore) (Ertl & Schuffenhauer (2009)), is calculated as a sum of of substructure-based fragment penalties and a molecular complexity penalty, obtained via chemical heuristics and a statistical analysis of PubChem compounds, which once again pose a huge risk of misleading out-of-domain results, giving low accessibility scores for hardly synthesizable polymers (Skoraczyński et al. (2023)).
- BRICS fragments decompose polymer repeat units incorrectly, while Bemis–Murcko scaffolds fail to distinguish polymerization motifs

These fragment- and scaffold-based approaches reveal fundamental domain mismatch issues, exemplified by alternative generative strategies. The PolyOne dataset used in polyBERT training (Kuenneth & Ramprasad (2023)) employs BRICS decomposition to fragment polymer repeat units into buildable blocks, yet this strategy, designed for small-molecule retrosynthesis, may incorrectly decompose polymers, yielding chemically implausible recombinations. Similarly, SMiPoly (Ohno et al. (2023)) uses rule-based synthesis templates to ensure synthetic accessibility, but it is restricted to the seven most popular polymer types, substantially limiting chemical diversity compared to unrestricted sequence-level generation. Although these approaches address specific objectives (synthetic feasibility), they sacrifice the chemical space breadth necessary for discovery.

These diverse approaches prioritize different design objectives, — synthetic accessibility, property targeting, or chemical diversity, — yet collectively lack a framework that combines high physicochemical fidelity with unrestricted chemical space exploration. Unconditional models like PI1M generate broadly but produce invalid or out-of-domain structures at high rates, while conditional and rule-based methods sacrifice diversity for constraints. This gap motivates PoGE: an unconditional generator that ensures both structural validity and alignment with real-world polymer distributions.

## 3 METHOD

### 3.1 POLYMER STRUCTURES DATASETS

Usually, when one discusses the validity of generated polymer structures, it usually limits by "chemically-valid SMILES" (RDKit could successfully process it), which is not enough for polymers.

Firstly, all the endpoints of the polymer repeat units should be connected only with one atom for the obvious chemical reasons. Moreover, all bonds, that connect atoms with the endpoints of the polymer repeat unit should be the same type (since they represent the same bond). Moreover, we considered only polymers with 2 endpoints of the polymer repeat unit ('*' symbol in SMILES) as valid since those structures form the vast majority in PolyInfo (98.2 %).

So, next we will consider the generated structure as polymer-valid ("p-valid", to avoid confusion with the well-established concept of "validity" in molecular cheminformatics) if the four following conditions are satisfied (Figure 2):

- Generated strings is a valid SMILES (passes RDKit SMILES-to-MOL conversion check)
- Every endpoint of the polymer repeat unit connects only with the one atom
- Each bond at the endpoints of the polymer repeat unit should be of the same type
- SMILES has 2 endpoints of the polymer repeat unit

As an example of real-world polymers, we used 18000 unique structures from the largest publicly available experimental database - PolyInfo (Otsuka et al. (2011)), which seems insufficient for training large-scale models like GPT.

Given our objective of developing a large language model capable of generating novel polymer architectures, we sought to compile a sufficiently large dataset of valid polymer SMILES strings. Currently, the most extensive corpus available is PI1M (Ma & Luo (2020)), comprising approximately one million SMILES representations. However, even this dataset may be inadequate for training deep neural networks effectively.

To augment the training data, we opted to generate additional plausible polymer structures using an alternative generative model. While no unconditional open-source polymer generation model

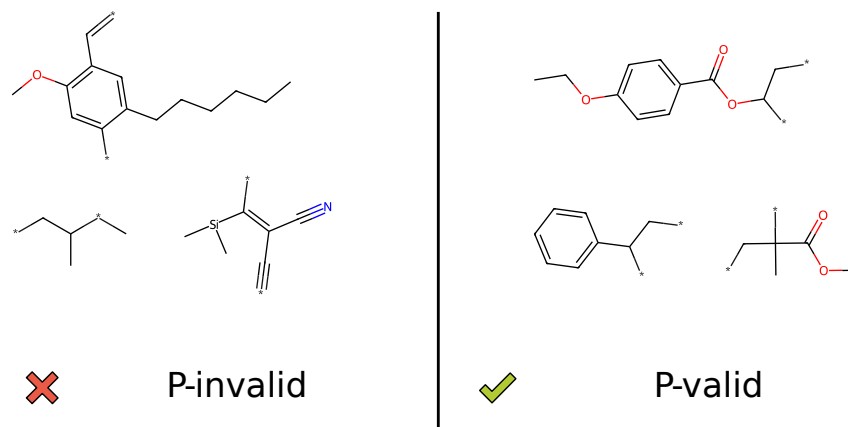

Figure 2: Examples of the p-valid and p-invalid polymer structures.

is currently available in the literature, we opted for PolyTAO (Qiu & Sun (2024)) as the closest accessible framework, a conditional polymer SMILES generator. Since PolyTAO requires a set of structural descriptors (e.g., molecular weight, number of rings, hydrogen bond donors, etc.) as input, we implemented an approach to approximate unconditional generation.

First, we computed all 15 relevant descriptors needed for PolyTAO input for each polymer in the PolyInfo dataset. Subsequently, we approximated the empirical distributions of these descriptor values. From these distributions, we sampled approximately one million sets of descriptors and used them as input for PolyTAO to generate a new dataset, hereafter referred to as the "PolyTAO corpus." See Appendix A.2.1 for more detailed description of the generation process.

The final training corpus was constructed by combining approximately one million structures from the PI1M dataset, one million structures from the PolyTAO corpus, and five times replicated of the PolyInfo dataset (i.e. duplicating PolyInfo five times in training corpora). Replicating higher-quality datasets during training is a well-established strategy in natural language processing for improving model performance on key data subsets (Brown et al. (2020)). Prior to replication, a validation split of 20% was reserved from the original dataset.

As the first part of data preprocessing procedure, all invalid SMILES were filtered out. For PolyInfo specifically, we identified and removed 0.1% invalid SMILES (structures that fail RDKit SMILES-to-MOL conversion) and 1.8% structures containing 3 or more endpoints. This filtering ensures alignment with the 2-endpoint constraint, which is consistent with PI1M's post-generation filtering criteria and reflects the dominance of linear homopolymers in the experimental literature. Surprisingly, PI1M which has been filtered before publication has 0.7% of invalid SMILES. Subsequently, only polymer structures deemed valid according to predefined structural and chemical criteria were retained for further use in model training. To facilitate efficient sequence modeling, all SMILES strings were tokenized using byte-pair encoding (BPE) (Sennrich et al. (2015)), a widely adopted technique in natural language processing. This approach serves two key purposes: first, it significantly reduces the total number of required tokens, which is particularly important given that RNN architectures such as Long Short-Term Memory (LSTM) typically struggle to retain information over sequences exceeding 5–7 tokens (Zhao et al. (2020); Al-Selwi et al. (2023)); second, it enhances the semantic coherence of individual tokens by grouping frequently occurring substructures (e.g., functional groups), thereby enabling the model to more effectively learn and reconstruct molecular architectures. Such SMILES-based transformers have already been successfully applied for conditional SMILES generation in the small-molecule domain (Wang et al. (2023); Mao et al. (2023); Tan (2025)).

## 3.2 MODEL ARCHITECTURE

While early polymer generation approaches - such as those employed in the PI1M dataset - relied on recurrent neural network (RNN) architectures, these models suffer from well-documented limi-

tations, including vanishing gradients and poor long-term memory retention. In contrast, we adopt a transformer-based architecture, specifically the GPT-2-like model, which leverages self-attention mechanisms to overcome these shortcomings. As a foundational member of the transformer family, GPT-2 enables improved contextual understanding and enhanced generation of structurally complex polymer structures.

Unlike RNNs, which encode sequential context into a single hidden state vector that is prone to information loss over long sequences, transformers maintain per-token hidden states through self-attention mechanisms. This allows the model to effectively capture long-range dependencies and generate structurally complex molecular representations without suffering from the vanishing memory limitations of RNNs (Zhao et al. (2020); Al-Selwi et al. (2023)).

The model architecture was deliberately scaled down to a reduced variant of GPT-2, featuring lower hidden dimensions (256 vs. 768 in the smallest GPT-2), fewer transformer layers (6 vs. 12), and fewer attention heads (8 vs. 12), due to the limitations in the quantity and quality of the available training data. Specifically, the PI1M dataset is limited by its size (one million structures) with no actual generative model provided, while the PolyTAO-generated data lack sufficient chemical diversity and contain inherent noise (Table 1), making it unsuitable as a main source of data for pre-training a larger model. Additionally, the PolyInfo dataset used for supervised fine-tuning is relatively small, and a larger model would risk overfitting during this phase, compromising generalization performance.

While transformers exhibit quadratic computational complexity with respect to sequence length (in contrast with the linear complexity of RNN), this limitation is mitigated in our application due to the relatively short length of polymer SMILES strings. Following tokenization via BPE, the average sequence length remains under 15 tokens, resulting in negligible computational overhead.

## 3.3 TRAINING PROCEDURE

Following standard practices in large language model training, the process was divided into two stages: pre-training and supervised fine-tuning (SFT). The pre-training stage utilized a large volume of typically lower-quality data, while SFT involved training on a smaller set of high-quality examples. For the pre-training phase, the previously described corpora — PI1M, the PolyTAO-generated dataset, and the replicated PolyInfo dataset — were combined. Subsequently, during the SFT phase, the model was further trained exclusively on the PolyInfo dataset. This strategy enabled the model to learn general structural patterns from the broader, noisier dataset during pre-training, followed by refinement of its generative capabilities using curated, experimentally validated polymer structures during supervised fine-tuning.

The generation parameters of the model were optimized using Optuna (Akiba et al. (2019)) to enhance model efficiency and ensure generation of chemically valid structures. The three hyperparameters optimized included temperature, top_p (nucleus sampling threshold), and top_k (number of highest-probability tokens considered during sampling), as they directly control generation diversity and validity. The optimization objective function was defined as the fraction of unique, p-valid polymers generated within a sample of 1 million SMILES strings. Hyperparameters were selected to directly optimize the target metric, ensuring that the configuration is aligned with domain requirements rather than generic language modeling objectives. Following 1000 iterations of optimization, the set of parameters delivering the highest performance was identified as temperature = 1.2145, top_p = 0.9848, and top_k = 398. This configuration achieved a maximum target fraction of 0.6515.

## 3.4 PERFORMANCE EVALUATION

**Validity** (valid) is calculated as a fraction of valid structures in generated set, using RDKit's molecular structure parser.

**Polymer-specific validity** (p-valid) is calculated as a fraction of valid polymers in generated set, using RDKit's molecular structure parser and criteria mentioned above.

**Uniqueness** (unique) shows a fraction of unique canonicalized SMILES for the first 100,000 valid polymers in the generated set.

**Novelty** (Novelty) is the fraction of the generated molecules that are not present in the experimental set (all PolyInfo data). This methodology was chosen specifically to avoid circular logic, as testing novelty against another generative model's output (e.g. PI1M or PolyTAO corpora) would measure mode coverage of previous synthetic generators rather than true chemical novelty

**Similarity to a nearest neighbor** (SNN) and **internal diversity** (IntDiv) are defined similarly to the MOSES original work (Polykovskiy et al. (2020)). SNN is computed as the average Tanimoto similarity $T(m_G, m_R)$ between the fingerprint of each polymer $m_G$ from the generated set $G$ and that of its nearest neighbor $m_R$ in the training dataset $R$:

$$SNN(G, R) = \frac{1}{|G|} \sum_{m_G \in G} \max_{m_R \in R} T(m_G, m_R) \tag{1}$$

IntDiv measures the chemical diversity among the molecules within the generated set $G$:

$$IntDiv(G) = 1 - \sqrt{\frac{1}{|G|^2} \sum_{m_1, m_2 \in G} T(m_1, m_2)} \tag{2}$$

For properties distributions comparison, we selected structural descriptors that meet rigorous criteria to ensure relevance and reliability in polymer analysis. All descriptors are directly computable from the polymer's atomic connectivity using RDKit, avoiding small-molecule-specific metrics like LogP (which relies on experimental partition coefficients) or SAScore (designed for synthetic accessibility of drug-like molecules). Instead, we prioritize features intrinsically tied to the polymer backbone and side-chain topology. Each descriptor is computationally efficient, relying solely on topological atom/bond counting or predefined contribution tables (for example, TPSA (Ertl et al. (2000)) through the RDKit atom-type-based algorithm), eliminating the need for resource-intensive quantum mechanical or molecular dynamics simulations.

These metrics were chosen for their chemical interpretability: molar mass reflects polymer size, rotatable bond fraction quantifies chain flexibility, heteroatom fraction indicates polarity and intermolecular interaction potential, aromatic fraction governs rigidity and $\pi$-stacking behavior, and TPSA serves as a standard proxy for surface polarity, allowing polymer chemists to directly relate numerical values to tangible structural and functional properties. Crucially, the descriptors exhibit minimal redundancy; for instance, the molar mass (total mass) is distinct from the heteroatom fraction (proportion of non-carbon atoms), and the aromatic fraction (proportion of aromatic atoms) is independent of the rotatable bond fraction (proportion of flexible bonds), avoiding multicollinearity that could obscure meaningful differences.

Finally, all descriptors except molar mass are normalized as dimensionless ratios per repeat unit (e.g., fraction of rotatable bonds, heteroatoms, or aromatic atoms), ensuring they remain invariant to chain length. This prevents misleading interpretations where artificially short repeat units might distort results. Molar mass is intentionally retained as a chain-length-dependent metric to validate consistent repeat unit generation, since incorrect or scaled-down repeat units would directly manifest in molar mass deviations, while the other descriptors remain stable and interpretable across polymer sizes.

The difference between these properties' distributions is estimated using Wasserstein distance between generated set $G$ with quantile function (inverse cumulative distribution function) $F_G^{-1}(t)$ and the training dataset $R$ with quantile function $F_R^{-1}(t)$.

$$W_1(G, R) = \int_0^1 |F_G^{-1}(t) - F_R^{-1}(t)| dt \tag{3}$$

All metrics are reported as mean ± standard deviation across 10 independent generation runs, each generating 100,000 polymers. For PI1M, metrics are reported as static values without error bars and do not include uniqueness and novelty, as the generative model, training dataset and code are unavailable.

## 4 EXPERIMENTS

The GPT-2 model (Section 3.2) was trained on all available datasets: the PolyTAO corpora (Section 3.1), PI1M Ma & Luo (2020), and PolyInfo Otsuka et al. (2011). The training process began with a pre-training stage, following the methodology outlined earlier.

Although attempts were made to fine-tune LLM trained on drug-like molecules, no success was reported; see the Appendix A.3 for further details.

### 4.1 RESULTS AND DISCUSSION

#### 4.1.1 GENERAL METRICS

Quantitative metrics for assessing the quality of generated structures can be broadly divided into two groups: structure-based (using SMILES or ECFP representations for evaluation) and descriptor-based. As structure-based metrics suitable for polymers, analogues of the evaluations from MOSES were chosen: validity, p-validity, uniqueness, novelty, similarity to nearest neighbor (SNN), and internal diversity (IntDiv). Our developed model for unconditional polymer SMILES generation outperforms previous approaches such as PI1M and PolyTAO. As shown in Table 1, PoGE achieves a validity score of $0.87 \pm 0.01$ and p-validity of $0.86 \pm 0.01$, surpassing PI1M[1] (validity: approx. 0.8, p-validity: N/A) and PolyTAO (validity: $0.71 \pm 0.03$, p-validity: $0.63 \pm 0.02$). Notably, PoGE generates $65 \pm 2\%$ unique polymers and exhibits superior similarity to real-world data, with an SNN score of $0.74 \pm 0.02$ — indicating closer alignment to the PolyInfo training set compared to PI1M (0.63) and PolyTAO ($0.31 \pm 0.04$).

Table 1: Structure-based metrics. Higher values indicate better performance.

| Model Name | Valid | p-valid | Unique | Novel | SNN | IntDiv |
|---|---|---|---|---|---|---|
| PI1M[1] | 0.80 | N/A | N/A | N/A | 0.63 | 0.86 |
| PolyTAO[2] | $0.71 \pm 0.03$ | $0.63 \pm 0.02$ | $0.63 \pm 0.02$ | $\mathbf{0.99 \pm 0.01}$ | $0.31 \pm 0.04$ | $\mathbf{0.89 \pm 0.03}$ |
| PoGE | $\mathbf{0.87 \pm 0.01}$ | $\mathbf{0.86 \pm 0.01}$ | $\mathbf{0.65 \pm 0.02}$ | $0.87 \pm 0.03$ | $\mathbf{0.74 \pm 0.02}$ | $0.87 \pm 0.02$ |

#### 4.1.2 FRAGMENT-BASED METRICS

However, traditional fragment-based metrics like BRICS fragments or Bemis–Murcko scaffolds yielded misleading results (Table 5). Due to fundamental differences in structural motifs between polymers and drug-like molecules, PolyTAO scored highest on these metrics despite poor alignment with PolyInfo. This discrepancy underscores the inapplicability of molecule-centric benchmarks to polymers, reinforcing the necessity of polymer-specific evaluation frameworks.

#### 4.1.3 DESCRIPTOR-BASED METRICS

Descriptor-based metrics (Table 2) provide further evidence of PoGE's effectiveness in capturing the statistical properties of real polymers. The Wasserstein distance, a metric derived from optimal transport theory, quantifies the minimal "cost" required to reshape the distribution of generated polymer descriptors to match the target distribution of real polymers. This cost is calculated as the total probability mass that must be relocated across the descriptor space multiplied by the distance each unit of mass travels, providing a geometrically sensitive measure of distributional similarity that accounts for both the magnitude and spatial relationships of differences between distributions. Unlike simpler divergence metrics that may fail when distributions have disjoint supports, the Wasserstein

---

[1]The original publication reports a validity score of 0.8 for PI1M. However, the absence of publicly available training and evaluation code compelled us to utilize a pre-filtered dataset provided by the authors. The metrics presented here may not accurately reflect the generator's true performance, as the preprocessing step likely excludes invalid or edge-case structures.

[2]As a conditional generator, PolyTAO's evaluation metrics primarily reflect pre-training data characteristics and are unsuitable for model comparison due to strong dependence on the conditioning method.

distance offers a continuous, interpretable evaluation of how closely the generated structures align with the statistical characteristics of real-world polymer datasets.

For molar mass, PoGE achieves a Wasserstein distance of $33 \pm 2$, significantly lower than PI1M (64) and PolyTAO ($195 \pm 24$), indicating a closer match to the empirical distribution observed in PolyInfo (Figure 3a). This metric reflects PoGE's ability to generate polymers across a realistic range of molecular weights, avoiding extremes that deviate from experimentally validated structures. Similarly, when evaluating properties like aromatic fraction, rotatable bond fraction (Figure 3b), heteroatom content, and topological polar surface area (TPSA), (Ertl et al. (2000)) PoGE consistently shows low Wasserstein distance values. These values highlight the model's capacity to replicate key physicochemical characteristics of real polymers, such as chain flexibility (via rotatable bonds) and polarity (via TPSA), which are critical for functional applications.

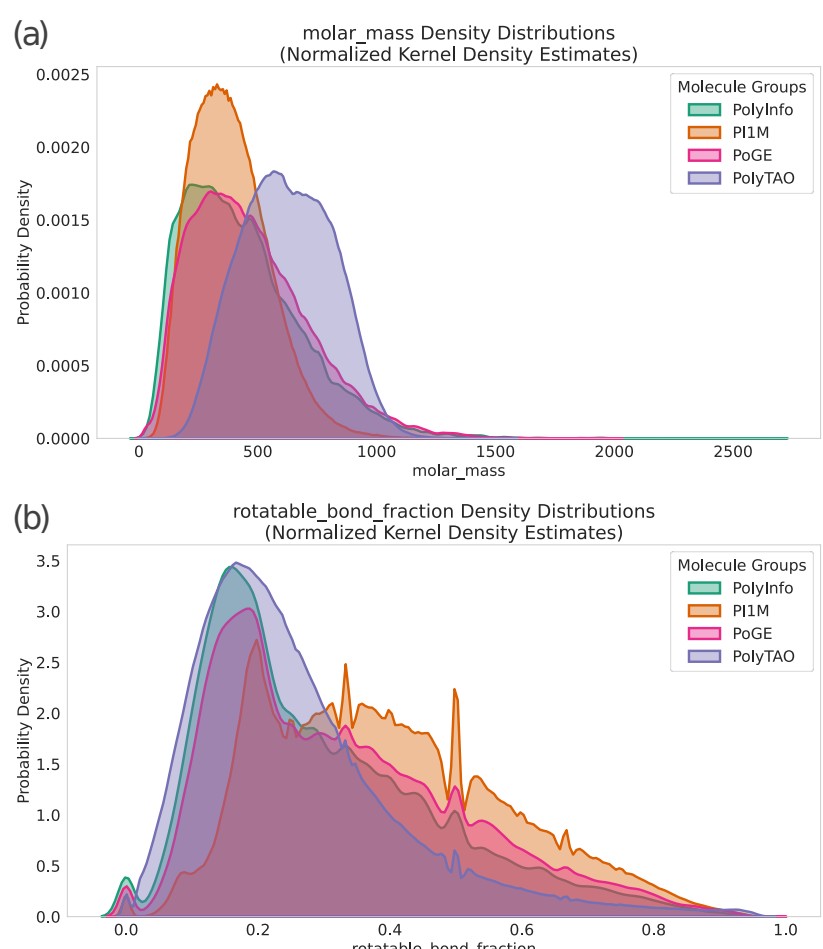

Figure 3: Normalized kernel density estimations of distribution of (a) – molar_mass and (b) – rotable_bond_fraction descriptors over the PolyInfo (red), PoGE (green), PolyTAO (cyan) and PI1M (purple)

Our work demonstrates the potential of adapting transformer-based architectures for polymer design, addressing existing challenges in both structural fidelity and evaluation rigor. By integrating polymer-specific validity criteria during training data selection and hyperparameter optimization, PoGE generates chemically coherent structures that align closely with experimentally validated data, as evidenced by its Wasserstein distances for molar mass, rotatable bond fraction and other distributions—a significant improvement over prior frameworks like PI1M and PolyTAO. Unlike traditional molecular generation models optimized for small molecules, PoGE's training strategy—combining large-scale synthetic corpora (PI1M and PolyTAO) with curated experimental data

Table 2: Descriptors-distribution-based metrics. (Wasserstein distance)

| Model Name | Molar mass | Aromatic fraction | Rotatable bond fraction | Heteroatom fraction | TPSA |
|---|---|---|---|---|---|
| PI1M | 64 | 0.068 | 0.062 | **0.010** | 8.3 |
| PolyTAO[1] | $195 \pm 24$ | $0.108 \pm 0.012$ | $0.038 \pm 0.005$ | $0.048 \pm 0.004$ | $17.5 \pm 0.8$ |
| PoGE | $\mathbf{33 \pm 2}$ | $\mathbf{0.026 \pm 0.002}$ | $\mathbf{0.030 \pm 0.002}$ | $0.016 \pm 0.001$ | $\mathbf{4.6 \pm 0.3}$ |

(PolyInfo)—enables it to capture key physicochemical properties, such as aromaticity and chain flexibility, without explicit descriptor conditioning. This outcome highlights the value of sequence-level modeling in learning implicit structural rules that govern polymer (or any other small-domain) chemistry.

A notable finding is that PoGE, a text-based generative model trained primarily on generating the p-SMILES, text molecule representations, achieves such fidelity in replicating PolyInfo's physical descriptor distributions. This outcome is particularly striking when compared to PolyTAO, a conditional generator explicitly designed to incorporate structural descriptors like molecular weight and hydrogen bond counts as input constraints. To approximate unconditional generation for PolyTAO, we sampled descriptor combinations from PolyInfo's empirical distributions and used them to guide its polymer generation pipeline, creating the PolyTAO corpus. Despite this tailored approach, PoGE surpasses PolyTAO in matching descriptor distributions, suggesting that the transformer architecture's NLP-inspired distribution alignment allows it to learn implicit structural rules from SMILES sequences more effectively than explicit descriptor conditioning.

## 4.2 LIMITATIONS

Despite its advancements, PoGE faces practical challenges that needs further investigation. Current synthetic accessibility metrics, such as SAscore, remain poorly suited for polymers, often penalizing large or repeating polymer structures and not taking into account common polymer-specific synthetic procedures. The development of polymer-aware scoring functions, however, remains critical for refining generative models, as they could enable more effective filtering of chemically plausible candidates or guide reinforcement learning strategies toward industrially relevant designs. Future work will prioritize developing polymer-aware scoring functions that account for repeat unit validity and industrial scalability.

## 5 CONCLUSIONS

The limitations of existing benchmarks in polymer evaluation underscore the critical need for domain-specific methodologies. Well-established tools like MOSES, which rely on fragment- or scaffold-based metrics designed for drug-like molecules, produce misleading results when applied to polymers due to fundamental structural differences. For instance, BRICS fragments incorrectly decompose polymer repeat units, while Bemis–Murcko scaffolds fail to distinguish polymerization motifs, leading to false assessments of scaffold diversity and chemical plausibility. To address these gaps, we introduced a physics-driven evaluation framework leveraging Wasserstein distance between descriptor distributions, focusing on interpretable physical and chemical properties such as molar mass, rotatable bond fraction, and topological polar surface area. This approach provides a physically grounded assessment of polymer generation quality, with our model achieving a molar mass Wasserstein distance of $33 \pm 2$ vs 64 for PI1M, demonstrating significantly better alignment with experimental data.

The introduction of PoGE addresses a critical gap in polymer informatics: the scarcity of open-access, large-scale datasets for pre-training. By releasing a comprehensive benchmark, high-quality pre-training corpus, and evaluation tools alongside the model, we establish a foundational platform

---

[1]As a conditional generator, PolyTAO's evaluation metrics primarily reflect pre-training data characteristics and are unsuitable for model comparison due to strong dependence on the conditioning method.

for conditional polymer generation tasks, including on-demand reverse design for specific applications like flexible electronics or gas separation membranes. This contribution extends beyond the model itself, enabling systematic comparisons of generative frameworks and fostering reproducible, domain-aware polymer discovery. Our work demonstrates that sequence-level modeling of polymer SMILES can implicitly capture complex physicochemical rules without explicit descriptor conditioning, paving the way for more effective AI-driven materials design in the polymer domain.

ETHICS STATEMENT

This work introduces PoGE (Polymer Generation and Evaluation), a framework for generating chemically valid polymers with accurate physicochemical properties. By enabling more accurate and reliable polymer design, our approach has the potential to accelerate the discovery of sustainable materials for critical applications such as renewable energy systems, biodegradable plastics, and advanced filtration membranes. These contributions could significantly reduce environmental impact and support the development of circular economy solutions.

However, as with any generative materials design tool, there are potential dual-use concerns. The framework could theoretically be applied to design novel polymers for harmful applications, such as advanced weapon systems or persistent environmental pollutants. To mitigate these risks, we emphasize the importance of ethical guidelines and oversight mechanisms for materials discovery research. We encourage researchers to consider the potential societal impacts of their work and to collaborate with policymakers to establish responsible innovation frameworks.

By releasing our model, training data, and evaluation tools as open-source resources, we promote transparency, reproducibility, and collaborative progress in polymer informatics. This approach aligns with the broader scientific community's commitment to responsible research practices and open science. We also note that our transformer-based architecture is designed for efficiency, requiring fewer computational resources than many alternative approaches, which helps minimize the environmental footprint of AI-driven materials discovery.

REPRODUCIBILITY STATEMENT

The authors fully support and advocate the principles of open science and reproducible research. All algorithms, architectures, and methodologies proposed in this work are open-sourced in a public code repository containing comprehensive training code, the trained PoGE model, 10 million generated polymer structures in CSV format, and a benchmark for performance evaluation of generative polymer models. The paper includes detailed descriptions of the model architecture (GPT-2 variant with 6 transformer layers, 256 hidden dimensions, and 8 attention heads) and critical hyperparameters (temperature = 1.2145, top_p = 0.9848, top_k = 398) used for generation. All visualizations presented in the main paper (including UMAP projections and descriptor distribution plots) and supplementary material explicitly document dataset selection criteria and specific examples used. Additionally, detailed experimental results are provided in the supplementary appendices, ensuring complete reproducibility of all results discussed in this work.

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

# A    APPENDIX

## A.1    KEY CHEMINFORMATICS CONCEPTS

In cheminformatics, molecular structures are represented and analyzed using specialized representations and heuristics designed to capture chemical relevance. We briefly define the key concepts used in this work:

- Extended Connectivity Fingerprints (ECFP) Rogers & Hahn (2010): ECFP is a widely adopted method for generating fixed-length binary vectors that encode the local chemical environment around each atom in a molecule. Starting from atomic properties (e.g., element type, degree, charge), ECFP iteratively expands the neighborhood of each atom up to a specified radius (typically 2–3 bonds), hashing combinations of atom environments into bit vectors. These fingerprints serve as structural descriptors that capture topological similarity and are commonly used as input features in ML models for property prediction, virtual screening, and molecular generation.

- RDKit's SMILES-to-MOL Conversion for Validity Assessments: SMILES (Simplified Molecular Input Line Entry System) is a string-based notation for representing molecular structures. RDKit is a popular open-source cheminformatics toolkit that parses SMILES strings into structured 3D-aware molecular graphs (MOL objects). In generative modeling, generated SMILES strings are often invalid due to syntax errors or unphysical bonding (e.g., pentavalent carbon). By attempting conversion to MOL format, we assess chemical validity: only molecules that successfully parse into a chemically plausible structure are considered valid outputs. The ratio of valid molecules is a standard metric for evaluating the quality of molecular generative models.

- BRICS Fragments Degen et al. (2008): BRICS (Breakable Rules for Intelligent Chemical Synthesis) is a rule-based fragmentation algorithm that decomposes molecules at pre-defined, synthetically plausible bond types (e.g., amide, ester, ether linkages). Each fragment retains chemically meaningful substructures that correspond to common small-molecule synthetic building blocks. BRICS fragments are used here to analyze modularity and reusability of generated molecules, enabling comparisons with known retrosynthetic pathways and facilitating interpretable analysis of scaffold diversity.

- Bemis–Murcko Scaffolds Bemis & Murcko (1996): A Bemis–Murcko scaffold is the core ring system and connecting linkers of a molecule, with all side chains removed. It captures the essential topological framework of a compound and is used to classify molecules by structural class. It is usually analyzed as a metric for avoiding repetitive or overly similar outputs in generative models, and for assessing coverage of chemically relevant chemical space for the domain of small molecules.

## A.2    TRAINING DATA ACQUISITION

### A.2.1    POLYTAO CORPUS GENERATION

PolyTAO is designed as a conditional generative model for polymer structures, requiring specific descriptor values (Table 3) as input for structure generation. To enable unconditional generation - that is, the generation of polymer structures without predefined constraints - the model must first be adapted to accommodate this objective.

To enable unconditional generation of polymer structures using the PolyTAO model, we adopted a two-step approach: (1) generating plausible descriptor values as inputs, and (2) obtaining polymer SMILES by feeding these values into the model.

Ensuring that the generated descriptor values accurately represent the real polymer population was critical. To achieve this, we derived empirical distributions for each descriptor from PolyInfo - the largest available database of experimentally validated polymers (Figure 4). For descriptors with discrete (natural number) values, we modeled their distributions using categorical distributions. The sole continuous descriptor, molecular weight (MolWt), was instead modeled using a log-normal distribution, given its strictly positive and right-skewed nature (Figure 5). The parameters of these

Table 3: PolyTAO input descriptors

| Descriptor Name | Descriptor detailed name | Descriptor type |
| --- | --- | --- |
| MolWt | Molecular weight of monomer | real number |
| HeavyAtomCount | Number of Heavy atoms in monomer | natural number |
| NHOHCount | Number of NHs or OHs in monomer | natural number |
| NOCount | Number of Nitrogens and Oxygens in monomer | natural number |
| NumAliphaticCarbocycles | Number of aliphatic carbocycles in monomer | natural number |
| NumAliphaticHeterocycles | Number of aliphatic heterocycles in monomer | natural number |
| NumAliphaticRings | Number of aliphatic rings | natural number |
| NumAromaticCarbocycles | Number of aromatic carbocycles in monomer | natural number |
| NumAromaticHeterocycles | Number of aromatic heterocycles in monomer | natural number |
| NumAromaticRings | Number of aromatic rings | natural number |
| NumHAcceptors | Number of Hydrogen Bond Acceptors in monomer | natural number |
| NumHDonors | Number of Hydrogen Bond Donors in monomer | natural number |
| NumHeteroatoms | Number of heteroatoms in monomer | natural number |
| NumRotatableBonds | Number of Rotatable Bonds in monomer | natural number |
| RingCount | Number of rings in monomer | natural number |

distributions were estimated via maximum likelihood estimation (MLE), ensuring a statistically robust fit to the observed data.

Following distribution parameter estimation, descriptor values were independently sampled from their marginal distributions to form inputs for the PolyTAO model. It is important to note, that we intentionally use independent sampling to avoid the risk of exposing explicit data leakage in the case of providing the PolyTAO model full parameters on joint distribution. This preserves the validity of using PolyTAO as pretraining data while respecting its design as a conditional, not unconditional, generator.

These inputs were then fed into the model to generate corresponding polymer SMILES strings. The sampling and generation process was iteratively repeated until a dataset of 1 million unique, valid polymer structures was obtained.

### A.2.2 DATA FILTRATION

We define the improved validity metric for the polymer structure (3.1). All SMILES strings used for model training satisfy described criteria.

### A.3 SMILES GENERATORS FINE-TUNING

Given the existence of pre-trained models for de-novo SMILES generation, which are trained on large datasets of bioactive molecules (e.g., ZINC, ChEMBL), a logical next step is to tune them for generating p-SMILES. Consequently, we selected SMILES-GPT (Adilov (2021)), a model based on the GPT-2 architecture pre-trained on the PubChem dataset, as our foundation model.

The primary objective of this generator is to propose novel polymer structures; therefore, its performance is critically dependent on the generation of unique and valid p-SMILES strings. Thus, the key evaluation metrics are the ratio of valid p-SMILES and the ratio of unique valid p-SMILES within the generated set.

Our initial approach involved a full fine-tuning of the SMILES-GPT model on a dataset of PolyInfo SMILES (which contains about 18k p-SMILES). This strategy yielded a notably low proportion of unique valid SMILES (0.2%), which was deemed insufficient for our purposes. Next, we employed Low-Rank Adaptation (LoRA) (Hu et al. (2022)) to train adapters specifically for the c_attn and c_proj attention layers. This approach, however, resulted in further performance degradation.

We hypothesized that this failure could be attributed to the low frequency of the '*' symbol in the model's pre-training corpus. This symbol is essential in p-SMILES notation to denote the two

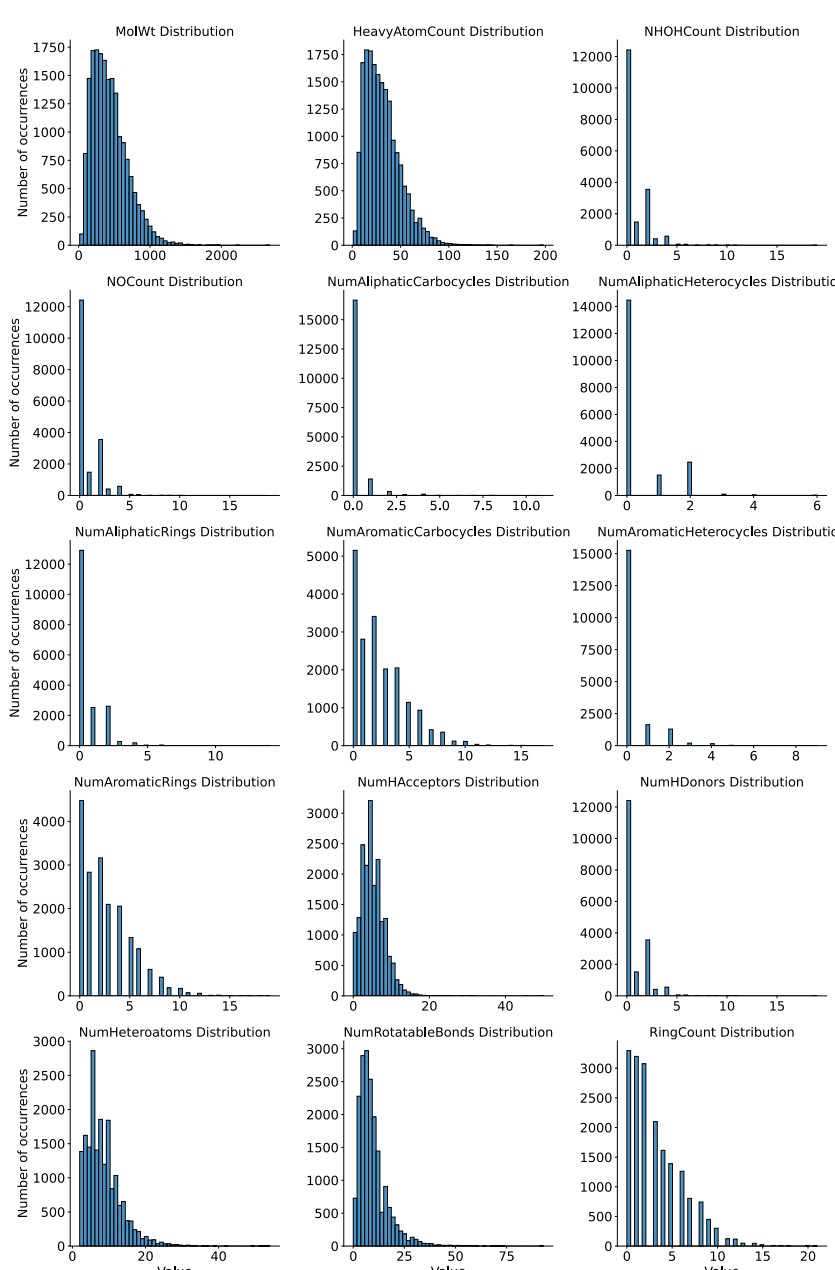

Figure 4: Distribution of PolyTAO descriptors in PolyInfo.

endpoints of the polymer repeat unit but is rare in standard small-molecule SMILES. To address this potential bottleneck, we expanded the LoRA training to include the token embedding (wte) and language model head (lm_head) layers. Unfortunately, this modification did not lead to any improvement in performance.

A final experiment involved applying the same LoRA methodology using the larger PI1M corpus (about 1m p-SMILES). This approach proved significantly more successful, producing 37.6% unique p-valid p-SMILES. However, this model was outperformed by the PoGE model across all descriptor-based distribution metrics (see Appendix A.4 for details).

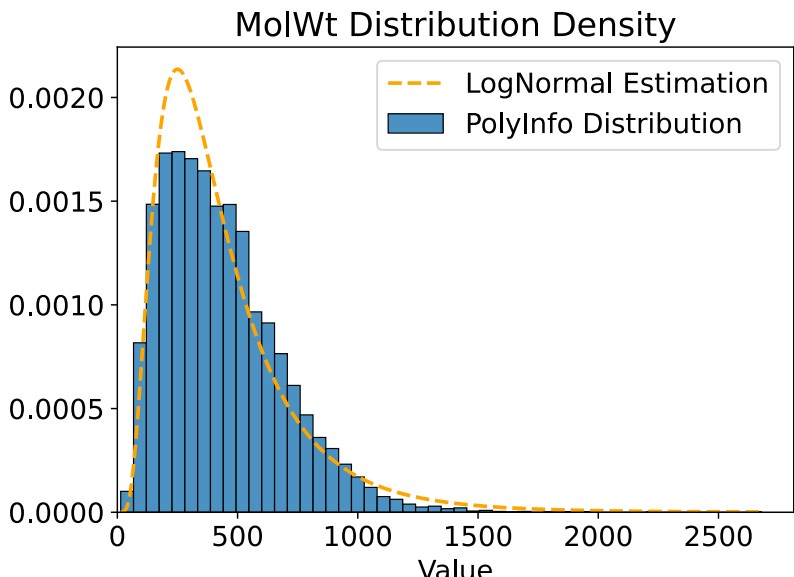

Figure 5: MolWt descriptor distribution from PolyInfo (blue bins) and its log-normal approximation (orange dashed line).

Table 4: SMILES-GPT fine-tuning results

| Model Description | Valid | p-valid | Unique valid | Unique p-valid |
|---|---|---|---|---|
| Full fine-tune on PolyInfo | 0.150 | 0.002 | 0.150 | 0.002 |
| LoRA for c_proj and c_attn layers on PolyInfo | 0.068 | 0.008 | 0.068 | 0.008 |
| LoRA for c_proj, c_attn, lm_head and wte layers on PolyInfo | 0.080 | 0.025 | 0.080 | 0.025 |
| LoRA for c_proj and c_attn layers on PI1M | 0.411 | 0.376 | 0.410 | 0.376 |
| LoRA for c_proj, c_attn, lm_head and wte layers on PI1M | 0.020 | 0.018 | 0.020 | 0.018 |
| Pretrain | 0.833 | 0.830 | **0.830** | **0.828** |
| SFT | **0.871** | **0.864** | 0.652 | 0.647 |

A comprehensive summary of all fine-tuning experiments, including benchmark scores from the pre-trained PoGE model and its supervised fine-tuned (SFT) version, is presented in Table 4. It is worth noting that PoGE after pre-training stage produces significantly more unique polymers than PoGE after SFT, however, its distribution-based metrics are far from optimal 7.

A.4 EXTENDED MODEL TESTING

UMAP projections (McInnes et al. (2020)) of ECFP representations (Figure 6) reveal that PoGE populates regions adjacent to PolyInfo clusters, implying synthesizability via known pathways. In contrast, PolyTAO and PI1M exhibit fragmented distributions, reflecting their reliance on conditional generation and smaller training sets, respectively. These results highlight PoGE's ability to produce diverse, chemically coherent structures.

Metrics based on BRICS fragments and Bemis–Murcko scaffolds are shown in Table 5.

Fragment similarity (BRICS) compares distributions of BRICS fragments in generated and reference sets. Denoting $c_f(A)$ a number of times a substructure $f$ appears in molecules from set $A$, and a set of fragments that appear in either $G$ or $R$ as $F$, the metric is defined as a cosine similarity:

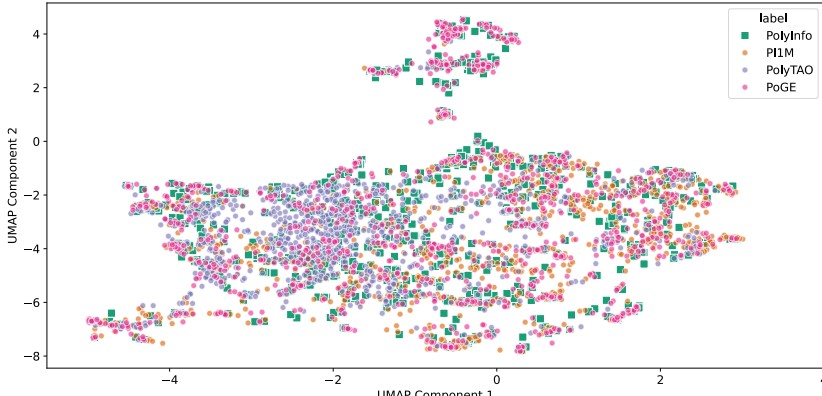

Figure 6: UMAP of ECFP representations of 1k samples of PolyInfo (green), PI1M (orange), Poly-TAO (purple) and PoGE (pink)

$$Frag(G, R) = \frac{\sum_{f \in F}[c_f(G) \cdot c_f(R)]}{\sqrt{\sum_{f \in F} c_f^2(G)} \cdot \sqrt{\sum_{f \in F} c_f^2(R)}} \quad (4)$$

Scaffold similarity (Bemis–Murcko scaffolds) is similar to fragment similarity metric, but instead of fragments we compare frequencies of Bemis–Murcko scaffolds. We use MOSES implementation of this algorithm which additionally considers carbonyl groups attached to rings as part of a scaffold. Denoting $c_s(A)$ a number of times a scaffold $s$ appears in molecules from set $A$, and a set of fragments that appear in either $G$ or $R$ as $S$, the metric is defined as a cosine similarity:

$$Frag(G, R) = \frac{\sum_{s \in S}[c_s(G) \cdot c_s(R)]}{\sqrt{\sum_{s \in S} c_s^2(G)} \cdot \sqrt{\sum_{s \in S} c_s^2(R)}} \quad (5)$$

While a comparison of the BRICS and Bemis–Murcko scaffolds similarities is provided in Table 5, its utility is limited. For instance, PolyTAO achieves the best scores in BRICS and Bemis–Murcko scaffolds yet performs worst in the SNN metric against the PolyInfo dataset. This discrepancy suggests that the BRICS and Bemis–Murcko scaffold fragments, which are derived from common synthons in organic chemistry, may not be appropriate for evaluating polymer datasets, where the relevant chemical building blocks differ significantly.

Table 5: BRICS and scaffold similarities

| Model Name | BRICS | Bemis–Murcko scaffolds |
|---|---|---|
| PI1M | 0.009 | 0.201 |
| PolyTAO | **0.107 ± 0.014** | **0.460 ± 0.038** |
| PoGE | 0.004 ± 0.001 | 0.058 ± 0.011 |

As it was mentioned previously, distribuition-based-metrics were also computed via Jensen–Shannon divergence (Table 6). Moreover, here one could find the extended table with distribution-based-metrics computed via Wasserstein distance for the best models from Appendix A.3 (Table 7).

Table 6: Extended descriptors-distribution-based metrics. (Jensen–Shannon divergence)

| Model Name | Molar mass | Aromatic fraction | Rotatable bond fraction | Heteroatom fraction | TPSA |
|---|---|---|---|---|---|
| PI1M | $0.150 \pm 0.021$ | $0.115 \pm 0.018$ | $0.132 \pm 0.022$ | $\mathbf{0.043} \pm 0.007$ | $0.089 \pm 0.016$ |
| PolyTAO | $0.359 \pm 0.061$ | $0.268 \pm 0.047$ | $0.104 \pm 0.018$ | $0.172 \pm 0.031$ | $0.148 \pm 0.025$ |
| PoGE (after SFT stage) | $\mathbf{0.060} \pm 0.007$ | $\mathbf{0.076} \pm 0.009$ | $\mathbf{0.062} \pm 0.008$ | $0.045 \pm 0.007$ | $\mathbf{0.038} \pm 0.005$ |
| PoGE (after pretrain stage) | $0.158 \pm 0.028$ | $0.272 \pm 0.046$ | $0.189 \pm 0.032$ | $0.050 \pm 0.009$ | $0.089 \pm 0.016$ |
| SMILES-GPT LoRA for c_attn and c_proj on PI1M | $0.180 \pm 0.032$ | $0.351 \pm 0.063$ | $0.288 \pm 0.046$ | $0.123 \pm 0.021$ | $0.124 \pm 0.021$ |

Table 7: Extened descriptors-distribution-based metrics. (Wasserstein distance)

| Model Name | Molar mass | Aromatic fraction | Rotatable bond fraction | Heteroatom fraction | TPSA |
|---|---|---|---|---|---|
| PI1M | 64 | 0.068 | 0.062 | $\mathbf{0.010}$ | 8.3 |
| PolyTAO | $195 \pm 24$ | $0.108 \pm 0.012$ | $0.038 \pm 0.005$ | $0.048 \pm 0.004$ | $17.5 \pm 0.8$ |
| PoGE (after SFT stage) | $\mathbf{33 \pm 2}$ | $\mathbf{0.026 \pm 0.002}$ | $\mathbf{0.030 \pm 0.002}$ | $0.016 \pm 0.001$ | $\mathbf{4.6 \pm 0.3}$ |
| PoGE (after pretrain stage) | $77 \pm 8$ | $0.128 \pm 0.020$ | $0.096 \pm 0.006$ | $0.011 \pm 0.002$ | $8.9 \pm 0.5$ |
| SMILES-GPT LoRA for c_attn and c_proj on PI1M | $120 \pm 15$ | $0.160 \pm 0.022$ | $0.118 \pm 0.016$ | $0.029 \pm 0.002$ | $19.3 \pm 1.2$ |