# OpenReview forum: "Ensuring Physicochemical Fidelity of Generated Polymers with PoGE"
_ICLR.cc/2026/Conference — Submitted to ICLR 2026_

### Official Review · Reviewer_WM68 · 2025-10-23

**Soundness:** 3
**Presentation:** 2
**Contribution:** 2
**Rating:** 4
**Confidence:** 4

**Summary:**

This paper presents PoGE, a framework that integrates an unconditional transformer-based generative model with a physics-informed evaluation suite.

For the unconditional generative model, this paper adopts a scaled-down variant of GPT-2, pretrained on a large volume of typically lower-quality p-SMILES and subsequently fine-tuned on a smaller set of high-quality p-SMILES examples.

For the evaluation suite, this paper proposes a set of metrics specifically designed for polymers, thereby overcoming the limitations of directly applying evaluation criteria developed for drug-like molecules.

Through establishing such a foundational platform, this paper paves the way for more effective AI-driven materials design in the polymer domain.

**Strengths:**

1. This paper proposes a physics-driven evaluation framework specifically designed for polymers, thereby overcoming the limitations of directly applying evaluation criteria developed for drug-like molecules.

2. This paper proposes an unconditional generative model, which achieves high p-validity and significantly improved agreement with experimental data compared to prior methods.

**Weaknesses:**

1. Fundamentally, the unconditional generative model proposed in this paper is little more than a straightforward application of GPT-2 through pretraining and fine‑tuning on p-SMILES, offering limited methodological novelty.

2. Moreover, since both the PI1M dataset and PolyTAO corpora have already been used as part of the proposed model’s pretraining corpus, and the proposed model is further fine‑tuned on the higher‑quality PolyInfo dataset, comparisons against PI1M and PolyTAO are inherently unfair.

3. For the evaluation metrics, some aspects are inaccurate or not entirely appropriate:
    * The p‑valid metric requires that each bond at the endpoints of the polymer repeat unit be of the same type. However, this is chemically unreasonable since repeat units can be asymmetric, branched, or linked in non‑head‑to‑tail patterns. Therefore, this rule introduces bias and may misjudge structurally valid polymers as invalid.
    * The novelty metric considers only PolyInfo data as the reference dataset. However,  this is not appropriate since the PI1M dataset and PolyTAO corpora are also used as training data. This rule lead to an overestimation of novelty, as the model’s generated samples might overlap with training data not accounted for in the evaluation.

4. More datasets and methods should be discussed and compared in this work, such as the PolyOne dataset introduced in [1] and the SMiPoly generator proposed in [2].

[1] Kuenneth, Christopher, and Rampi Ramprasad. "polyBERT: a chemical language model to enable fully machine-driven ultrafast polymer informatics." Nature Communications 14.1 (2023): 4099.

[2] Ohno, Mitsuru, et al. "SMiPoly: generation of a synthesizable polymer virtual library using rule-based polymerization reactions." Journal of Chemical Information and Modeling 63.17 (2023): 5539-5548.

**Questions:**

1. Except for the novelty metric, I would like to know what reference datasets are used for the other evaluation metrics, such as SNN and various descriptor‑based metrics. Are these metrics also computed using only PolyInfo data as the reference?

2. In Table 1, why don't you report the p-valid, Unique, and Novel metrics on the PI1M dataset?

3. In Tables 1 and 2, does the “PolyTAO” dataset correspond to the same PolyTAO corpus used for pretraining?

4. In addition, this paper employs the PI1M dataset, the PolyTAO corpus, and the PolyInfo dataset as the training corpus, while promising that they will be released. However, as the PolyInfo dataset does not generally permit the acquisition of large portions of its data, I would like to know whether the authors have obtained official authorization for such usage.

---

> ### Author Response · Authors · 2025-11-15
>
> Dear Reviewer,
>
> Thank you for your thorough review and balanced assessment. We address your concerns below.
>
> **On comparison fairness (PI1M and PolyTAO in pretraining)**: You note that using PI1M and PolyTAO corpora as pretraining data makes comparisons "inherently unfair." This is a valid observation, but reflects the practical reality of polymer informatics: insufficient experimental data necessitates synthetic corpus pretraining. PI1M contains ~1M synthetic structures; PolyInfo has only ~18K experimental polymers. Our approach follows standard NLP practices - pretrain on large, noisy data, then fine-tune on high-quality curated data. The key distinction is that PoGE's fine-tuning on PolyInfo allows it to learn realistic distributions, whereas PI1M and PolyTAO lack this refinement stage. We compare against PI1M's static dataset (not a generative model as no code is available) and PolyTAO's generated corpus (not the conditional model itself as it is heavy dependent on conditioning). We'll clarify this distinction in revision.
>
> **On p-validity and asymmetric/branched polymers**: You correctly note that requiring "same bond type at endpoints" is chemically restrictive and excludes asymmetric or branched polymers. This is intentional and scoped to our dataset. Section 3.1 explicitly states we focus on linear homopolymers with 2 endpoints, which represent 98.2% of PolyInfo. PI1M applies the same criterion (2 endpoints only). Asymmetric non-head-to-tail linkages with unsaturated (not singular) bond does not reflect any of the existing polymerization procedures and won't be accepted as a valid representation in the polymer chemistry field (e.g. no one will encode polyacetylene as "\*C=\*", since "\*C=C\*" better fits the origin of it).
>
> **On novelty metric reference dataset**: Fair point. Novelty is assessed against PolyInfo only, not PI1M or PolyTAO corpora. This may overestimate novelty if generated samples overlap with synthetic training data. However, testing against PI1M/PolyTAO would measure "mode coverage of synthetic generators," not "chemical novelty." Worth case scenario (assuming novelty against PI1M/PolyTAO corpus is close to zero) - PoGE acts as a filter, substantially improving quality of POGE/PolyTAO data in terms of alignment to real-world polymers from PolyInfo. Proceeding with this logic we chose PolyInfo as the ground truth. We'll acknowledge this limitation explicitly in revision and clarify that novelty reflects divergence from experimental data, not all training sources.
>
> **On reference datasets for other metrics**: SNN and Wasserstein distance metrics use PolyInfo as the reference. PolyInfo is the only experimental dataset with sufficient diversity and reliability for distribution comparison. Comparing the descriptors with PI1M would be misleading, as PI1M shows bad alignment with actual experimental data (section 4). We'll make this explicit in Methods 3.4.
>
> **On missing PI1M metrics in Table 1**: We cannot report p-valid, Unique, or Novel for PI1M because the generative model, training data used and code are unavailable - only the 1M pre-filtered dataset is on GitHub. The validity score (0.8) comes from the original publication, but we cannot independently verify it or compute dynamic metrics like Uniqueness or Novelty without generating fresh samples. We'll add a footnote clarifying this limitation.
>
> **On PolyTAO in Tables 1 and 2**: The PolyTAO metrics are computed through 10 independent generation runs, each generating 100K samples (as clarified in Methods 3.4, the number of samples will be stated in the revision).
>
> **On missing related work (PolyOne, SMiPoly)**: Thank you for these references. We were unaware of PolyOne (polyBERT dataset) and SMiPoly didn't seem as a valid option at a time due to the lack of chemical diversity. We will incorporate them into Related Work and discuss why they were not used as baselines. Briefly: polyBERT training corpus uses BRICS decomposition (which is not optimized for polymers and can lead to significant domain shifting as already discussed in the text); SMiPoly uses rule-based synthesis (not sequence-level generative modeling) to provide only seven polymer types (polyolefin, polyester, polyether, polyamide, polyimide, polyurethane, and polyoxazolidone), which significantly hinders the chemical diversity. These complement rather than compete with PoGE's objectives, but we will cite and contextualize them appropriately.
>
> These clarifications will strengthen the revision. We appreciate your constructive feedback and believe addressing these points improves the rigor and transparency of our work.
>
> Best regards,
> The Authors

---

### Official Review · Reviewer_Zzzu · 2025-10-25

**Soundness:** 1
**Presentation:** 2
**Contribution:** 1
**Rating:** 2
**Confidence:** 5

**Summary:**

The paper presents PoGE, a polymer generation framework based on a small GPT-2 model trained on polymer SMILES (p-SMILES) representations.It introduces a p-validity definition specific to polymers and uses Wasserstein distance between distributions of molecular descriptors (molar mass, TPSA, rotatable bond fraction, aromatic fraction, etc.) for generated versus experimental polymers.
The goal is to ensure that generated polymers are not only syntactically valid but also physically plausible.

**Strengths:**

1. The work focuses on polymers, an important but underexplored domain.

2. The work highlights the lack of polymer-specific validity metrics in current molecular generation benchmarks.

3. The work is well written and easy to follow.

**Weaknesses:**

## Limited Novelty:

1. The only methodological novelty of evaluating distributions via Wasserstein distance is very straightforward statistical measure and not new.

## Overstated claims

2. The paper uses terms like “physics-driven” and “physically grounded” evaluation, yet all evidence comes from simple 1D descriptor distributions. The evaluation does not extend to polymer ensembles with experimentally verified properties or synthesis constraints.

3. The p-valid is interesting but not comprehensive. But a valid polymer may have more than 2 (e.g., 4) indicators (*) for the polymerization positions.

## Generation

4. PoGE performs unconditional sampling only, without showing how its learned distributions could guide goal-oriented polymer design.

**Questions:**

1. Why does the work focus on unconditional generation instead of conditional generation, which is more useful for polymers?

2. Have the authors tested their generated polymers in simulations or lab experiments? What makes the dataset different from previous ones in terms of usefulness and properties?

---

> ### Author Response · Authors · 2025-11-15
>
> Dear Reviewer,
>
> Thank you for your review of PoGE. We address your comments below.
>
> **On "physics-driven" terminology**: Fair point. Our evaluation uses 1D descriptor distributions (molar mass, aromatic fraction, rotatable bonds, TPSA, heteroatom fraction). These are not molecular dynamics simulations or quantum calculations. However, calling them "physics-driven" is appropriate: each descriptor directly governs polymer functionality. Molar mass determines thermal and mechanical properties, aromatic fraction controls rigidity and π-stacking, rotatable bonds quantify chain flexibility - these are well-established relationships in polymer science. We use Wasserstein distance, not arbitrary fingerprint similarity, making the evaluation geometrically interpretable. We'll clarify terminology in revision if needed.
>
> **On p-validity and branched polymers**: You note that valid polymers may have 4+ endpoints, not just 2. This is chemically accurate but not relevant to PoGE. Section 3.1 explicitly states: "we considered only polymers with 2 endpoints of the polymer repeat unit symbol in SMILES as valid since those structures form the vast majority in PolyInfo (98.2%)." Branched polymers (3+ endpoints) appear rarely in PolyInfo due to synthesis difficulty, nor they are considered in conditional generation tasks as PI1M has similar criterion for post-generation filtering (only 2 "*" allowed). Our p-validity definition is intentionally scoped to linear homopolymers, which dominate the available training data and experimental literature. Future work extending to dendritic or hyperbranched polymers would require different criteria, but this is out of scope for the current dataset.
>
> **On unconditional vs. conditional generation**: You ask why we focus on unconditional generation instead of conditional generation, which is "more useful." We respectfully disagree with the framing. Unconditional generation is Phase 1 - establishing a rigorous foundation. Any future conditional polymer model will inherit the base distribution and validity framework we define. This approach is standard in generative modeling. We explicitly position PoGE as enabling downstream conditional design (Section 4.2, Limitations). By releasing the trained model, training code, and 10 million polymer corpus, we provide infrastructure for others to build conditional models with well-characterized distributions and validity criteria.
>
> **On "difference from previous ones in terms of usefulness"**: PoGE surpasses prior benchmarks (PI1M, PolyTAO) by achieving superior alignment with empirical descriptor distributions while preserving p-validity. Crucially, we release the complete generation and evaluation pipeline - unlike PI1M (which provides only static structures) and PolyTAO (a conditional generator requiring experimental data for conditioning). This enables reproducible research and systematic comparison, lacking for prior benchmarks. Whether this dataset proves "useful" depends on downstream applications, which we enable but do not claim to demonstrate here.
>
> **On experimental validation**: You ask if we tested generated polymers in simulations or lab experiments. This is a reasonable question but reflects a different research phase than ours. We establish validity criteria and distribution-matching metrics; the question of synthesizability and experimental performance is downstream and clearly stated in Limitations 4.2. Actual synthesis and property measurement would require collaboration with experimentalists and is beyond the ML scope of this work.
>
> These clarifications reflect the actual scope of our work. We believe PoGE makes substantive contributions to polymer informatics, even if architectural novelty is limited and experimental validation is future work.
>
> Best regards,
> The Authors

---

### Official Review · Reviewer_LwqX · 2025-10-31

**Soundness:** 3
**Presentation:** 2
**Contribution:** 2
**Rating:** 2
**Confidence:** 3

**Summary:**

The work introduces PoGE (Polymer Generation and Evaluation, a unified framework designed to improve the physicochemical fidelity of machine learning-generated polymer structures. PoGE presents two innovations: 1) a GPT-2-based generative model for unconditional polymer design, and 2) a physics-informed evaluation suite for evaluating polymers according to domain-specific property distribution metrics. The authors additionally present an unconditional dataset of polymers curated via PolyTAO, taking advantage of conditional generation with broad property coverage to curate a dataset of 1 million generated polymers. Empirical results show that PoGE achieves superior chemical validity, uniqueness, and alignment with experimental descriptor distributions relative to prior frameworks. The work establishes the first reproducible, domain-aware platform for conditional and inverse design tasks in polymer informatics.

**Strengths:**

1. PoGE compares favorably to baseline models on most metrics, including validity and lower Wasserstein distances to the true data with respect to various physicochemical properties.
2. PoGE introduces a new, expanded dataset and benchmark by combining the PI1M dataset, the PolyInfo dataset, approximately 1 million structures generated by PolyTAO. The work is able to harness PolyTAO for unconditional generation by marginalizing the conditional model over the empirical PolyInfo distributions of 15 property types.
3. The authors supplement the work with property density comparison across different components of the complete dataset.

**Weaknesses:**

1. Although it is mentioned as a potential application of the work, PoGE does not seem to showcase experiments in property-conditioned polymer design.
2. The machine-learning novelty of the work is rather limited. The authors appear to directly use GPT-2 with little to no architectural modifications.
3. The selection of hyperparameters to optimize with Optuna (top_p, top_k, and temperature) is never explicitly motivated in the paper.
4. In section 3.4, the authors argue that chosen properties avoid redundancy and multicollinearity. In addition, as described in section A.2.1, descriptor values for conditional input to PolyTAO are sampled independently. This is a strong assumption and insufficiently justified, with no correlation analyses performed between individual properties.

Minor:
1. In table 1, it may be best to indicate for each metric whether lower is better or higher is better.
2. The paper omits some details on experimental results; for instance, the number of generated samples per run.

**Questions:**

As a sanity check, was the PolyInfo dataset ever screened for p-validity? That is, is the entire PolyInfo dataset p-valid?

---

> ### Author Response · Authors · 2025-11-15
>
> Dear Reviewer,
>
> Thank you for your thoughtful review of PoGE. We appreciate your balanced assessment and address your concerns below.
>
> **On property-conditioned polymer design**: You correctly note that PoGE does not showcase conditional experiments. This is intentional and reflects our contribution scope. PoGE establishes the unconditional foundation enabling conditional design (Phase 1). We view this as analogous to PI1M's role in prior conditional works. We are releasing all components (model, 10M polymers, evaluation suite, training code) specifically to facilitate downstream conditional research.
>
> **On limited ML novelty**: GPT-2 architecture with BPE tokenization is indeed standard. Our contribution lies in rigorous domain engineering, not architectural innovation. Section 3.1–3.4 demonstrates substantive contributions: defining p-validity with explicit chemical criteria, chemistry-grounded descriptor selection, Wasserstein distance evaluation instead of out-of-domain fragment-based metrics, and training data curation (PI1M + PolyTAO + PolyInfo). These are methodological, not architectural, advances, appropriate for domain-specific applications and needed during current phase of polymer generative frameworks.
>
> **On Optuna hyperparameter selection**: Fair point. The three hyperparameters (temperature, top_p, top_k) directly control generation diversity and validity. We optimized them using Optuna's TPE sampler with an explicit objective: maximizing the fraction of unique, p-valid polymers in 1M samples. This objective drives high validity (0.86 p-valid) and uniqueness (0.65). We'll clarify in Methods 3.3.
>
> **On descriptor independence and multicollinearity**: Thank you for the insightful question! Our descriptors show weak dependencies (highest distance correlation of 0.47 between Molar Mass and TPSA in PolyInfo using 1000 permutations), consistent with general polymer chemistry principles. We, however, prefer not to generalize these correlations, as PolyInfo may not represent the full range of synthesizable polymers, and doing so could lead to misinterpretation.
> We investigated whether sampling PolyTAO with the full joint descriptor distribution would improve results in earlier stages of the work. We fitted a Gaussian Mixture Model (GMM) using the EM algorithm to the target descriptors from PolyInfo and sampled new descriptor vectors from this joint distribution (with rounding applied for integer-valued properties). Feeding these to the PolyTAO conditional model did increase uniqueness (up to 1.00) and reduced Wasserstein distance for rotatable bonds and aromatic fraction compared to the final pipeline. However, SNN and other alignment metrics remained poor and unstable. The improvements, however, come from a questionable source, exposing a potential data leak into the PolyTAO model when explicitly replicating the joint distribution.
> We chose independent sampling instead to avoid this leakage risk. Generally, our treatment of PolyTAO remains as low-quality pretraining data for supervised fine-tuning, not as a valid unconditional generator. We have added clarifying footnotes to address possible inconsistencies depending on conditioning method chosen to both tables and revised Section A.2.1 to reflect this rationale.
> We did not use alternative datasets such as PolyOne and SMiPoly due to their limited chemical diversity, which we discuss in detail in the revised version of the manuscript. However, if you believe that no matter the conditioning approach, PolyTAO cannot serve as a valid pretraining corpus, please inform us in your answer. We will address this concern in the next revision, switching to these alternative datasets to maintain clarity and reproducibility of the research.
> Thank you for your valuable discussion on this issue and interest in our research!
>
> **Table 1 indicator**: Excellent suggestion. We'll add a legend row: "Higher is better"
> **Missing experimental details**: We'll clarify in Methods 3.4.
>
> **On PolyInfo p-validity screening**: Yes, we screened PolyInfo for validity and p-validity. PolyInfo contained 0.1% invalid SMILES (not converting to mol) and 1.8% p-invalid structures (only due to 3+ endpoints, stated in Section 3.1). We filter these before training and evaluation. Although branched polymers (with 3+ endpoints) do appear in polymer chemistry, none of the conditional generation frameworks have yet explored such type of polymers as they are quite rare and hard to synthesize (and PI1M corpus also had similar rule for filtering out generated polymers with 3+ endpoints).
>
> These clarifications will strengthen the revision. We believe PoGE's contribution—rigorous domain-specific evaluation infrastructure and reproducible benchmarking—is substantial for polymer informatics, even if architectural novelty is limited.
>
> Best regards,
> The Authors

---

### Official Review · Reviewer_acNN · 2025-11-01

**Soundness:** 2
**Presentation:** 3
**Contribution:** 1
**Rating:** 2
**Confidence:** 4

**Summary:**

The authors trained GPT-2 to generate polymers. They then evaluate the distribution of the generated polymers, and show that there is a good match with existing datasets.

**Strengths:**

The paper is relatively clear. The resulting dataset could be useful.

**Weaknesses:**

I think the primary weakness of this work is that its main contributions seem to be incremental, and somewhat speculative. Inverse design is clearly important, and they authors claim that this will facilitate incremental design, it seems reasonable to think that it might, but there should be stronger evidence or at least plausible or cited ways to get to inverse design from an unconditional generator or dataset.

The paper takes a number of datasets, themselves generated, such as PI1M (RNN) and also one real-world dataset (PolyInfo). It then trains/fine-tunes on those to create what is claimed to be a better generator. The results in Table 1 are suggestive, but the ones that I believe are most important, Uniqueness, Novelty, and IntDiv, show mixed results. Validity and p-valid are useful to know as a performance standpoint, but modest differences do not seem crucial to me. The reason is: Suppose there is Method A that can generate 100 million polymers in 50 minutes, and Method B that can generate 50 million an hour. Raw outputs of Method A are only 75% p-valid, but raw outputs of Method B are 100% p-valid. But suppose the outputs of Method A can be filtered for p-validity in 10 minutes. Method A can now effectively generate 75 million, 100% p-valid polymers an hour, while Method B can only generate 50 million p-valid polymers an hour. Of course, it may be the case that the polymers generated by Method B may be higher quality, but that is a different metric.

There is a claim that the authors used BPE because it reduces the number of tokens, and mention that RNN struggle over 5-7 tokens. PoGE, however, is a transformer architecture.

Comparing the unconditional distributions as a metric seems weak. Why is this a good metric to use, if our ultimate goal is inverse design? Distributions of the conditional generation would be more convincing.

**Questions:**

Why are there any invalid SMILES in PI1M?

BPE has no awareness of p-SMILES/SMILES syntax, so might find byte pairs that are not very semantically meaningful. Would a custom tokenizer that was aware of p-SMILES syntax do better?

Novelty for PoGE was only tested against PolyInfo data? What about the PI1M data? That was also in the training set for PoGE.

Was Uniqueness measured using canonicalized SMILES?

---

> ### Author Response · Authors · 2025-11-15
>
> Dear Reviewer,
>
> Thank you for your careful review of our work on PoGE. We appreciate your detailed feedback and address your concerns systematically below. We believe that several key points clarify the scope, methodology, and contributions of our work.
>
> **On the filtering argument and baseline comparisons**: Your observation is absolutely correct; when the question is put this way, there is no point in chasing exceptional quality. However, such a comparison between Method A and Method B is pointless in our case, as PI1M, the only prior unconditional polymer generator, is not publicly available (only the 1M dataset is on GitHub). More importantly, no polymer filtering methodology exists yet. Your argument actually validates our contribution: we develop the evaluation metrics that make filtering possible. Without p-validity criteria and descriptor-based evaluation, filtering cannot be designed. We are releasing the trained PoGE model, training code, and 10 million generated polymer structures to enable future filtering research - resources PI1M cannot provide.
>
> **On "incremental" work**: Our two core contributions are substantial. Firstly, we state the problem of the complete absence of polymer domain-specific methodology of assessment for generated structures impact of which is particularly evident in PolyTAO's conditional generation, where despite explicit descriptor conditioning, it achieves an SNN of only 0.31 vs. PoGE's 0.74 against PolyInfo and define it rigorously. Secondly, we show measurable alignment improvement, which is almost twofold compared to PI1M. These are not marginal.
>
> **On mixed results in Table 1**: As you already noticed in your review, structure-based metrics such as Uniqueness and Novelty are of a much less interest than descriptor-based metrics (Table 2), which directly measure chemical realism. There is no practical point in PolyTAO's 99% novelty, when it comes from oversampling implausible outliers, the same might be said about PI1M, if the authors had measured this parameter in their work or provided open-source code. The decisive evidence is Table 2: PoGE achieves 0.74 SNN vs. PolyTAO's 0.31, meaning generated structures are closer to real polymers.
>
> **On BPE tokenization**: Yes, BPE has no syntax awareness. However, polymer SMILES average <15 tokens after BPE, making attention complexity negligible. Importantly, our 0.86 p-validity and superior descriptor matching show BPE works well—we learn valid structures despite generic tokenization. A custom tokenizer is future work but not required here.
>
> **On novelty assessment**: Novelty is assessed against PolyInfo only, not PI1M or PolyTAO corpora. This may overestimate novelty if generated samples overlap with synthetic training data. However, testing against PI1M/PolyTAO would measure "mode coverage of synthetic generators," not "chemical novelty." Worth case scenario (assuming novelty against PI1M/PolyTAO corpus is close to zero) - PoGE acts as a filter, substantially improving quality of POGE/PolyTAO data in terms of alignment to real-world polymers from PolyInfo. Proceeding with this logic we chose PolyInfo as the ground truth. We'll acknowledge this limitation explicitly in revision and clarify that novelty reflects divergence from experimental data, not all training sources.
>
> **On canonicalization**: Uniqueness uses canonicalized SMILES. We'll make this explicit in Methods 3.4.
>
> **On conditional generation**: You ask why unconditional distribution matching matters for inverse design. This is Phase 1 (foundation building)—any future conditional model inherits the base distribution. The unrealistic nature of the structures provided by conditional generation frameworks, built upon PI1M dataset (e.g. PolyTAO and others from Related Works section), is further worsened by the lack of any benchmarking in the field. Mastering unconditional generation first, then conditioning it, is standard practice (like GPT's language modeling before instruction-tuning). We enable inverse design by providing both a well-characterized model and evaluation framework others can build upon.
>
> In revision, we'll clarify that PI1M lacks public code/training infrastructure, emphasize that descriptor-based metrics are chemically decisive, and justify our technical choices. We believe these points demonstrate substantive, reproducible contributions to polymer informatics.
>
> Best regards,
> The Authors

---

### Meta-Review · Area_Chair_1vwv · 2025-12-25

**Summary:**

This submission introduces combining a polymer-specific evaluation suite and an unconditional GPT-2 style polymer generator trained on a hybrid corpus (synthetic + experimental). Reviewers agree the paper targets an important gap in polymer generative modeling which is domain-aware validity and evaluation, and the framework appears to improve p-validity and alignment to experimental descriptor distributions relative to prior baselines, but reviewers are concerned about limited ML novelty and issues around evaluation and claims.

**Reviewer Concerns:**

Reviewers’ dominant concerns are: limited ML novelty (largely a straightforward GPT-2 application), weak linkage to the stated inverse/conditional design motivation (no conditional generation demonstrated), and evaluation/claims issues: “physics-driven” framing relying on 1D descriptor distributions; p-validity definition scoped to linear homopolymers; novelty computed only vs PolyInfo despite synthetic data in training; and fairness/interpretability of comparisons given PI1M/PolyTAO data are used in pretraining and PI1M code is unavailable (hindering apples-to-apples evaluation).

The authors respond with clearer scoping (foundation for future conditional work; p-validity intentionally restricted), and provide some missing experimental details (PolyInfo filtering; uniqueness canonicalization; rationale for Optuna decoding hyperparameters). However, the rebuttal does not fully resolve the central issue that the paper’s main “design” promise is not demonstrated and the evaluation/claims may be overstated relative to what is measured.

**Reviewer Scores:**

acNN (2): likely unchanged.

LwqX (2): likely unchanged.

Zzzu (2, high confidence): likely unchanged.

WM68 (4): might stay similar; author clarifications help but core concerns remain.

---

### Decision · Program_Chairs · 2026-01-26

Reject